# Estradiol Prevents Amyloid Beta-Induced Mitochondrial Dysfunction and Neurotoxicity in Alzheimer’s Disease via AMPK-Dependent Suppression of NF-κB Signaling

**DOI:** 10.3390/ijms26136203

**Published:** 2025-06-27

**Authors:** Pranav Mishra, Ehsan K. Esfahani, Paul Fernyhough, Benedict C. Albensi

**Affiliations:** 1Division of Neurodegenerative & Neurodevelopmental Disorders, St. Boniface Hospital Albrechtsen Research Centre, University of Manitoba, Winnipeg, MB R2H 2A6, Canada; pmishra@sbrc.ca (P.M.); eesfahani@sbrc.ca (E.K.E.); pfernyhough@sbrc.ca (P.F.); 2Department of Pharmacology and Therapeutics, Max Rady College of Medicine, Rady Faculty of Health Sciences, University of Manitoba, Winnipeg, MB R2H 2A6, Canada; 3Department of Pharmaceutical Sciences, Barry & Judy Silverman College of Pharmacy, Nova Southeastern University, Fort Lauderdale, FL 33328, USA

**Keywords:** Alzheimer’s disease, Amyloid-β, AMPK, estradiol, mitochondria, neuroprotection, neuroinflammation, NF-κB

## Abstract

Alzheimer’s disease (AD), the most common form of dementia, is a progressive neurodegenerative disorder characterized by memory loss and cognitive decline. In addition to its two major pathological hallmarks, extracellular amyloid beta (Aβ) plaques and intracellular neurofibrillary tangles (NFTs), recent evidence highlights the critical roles of mitochondrial dysfunction and neuroinflammation in disease progression. Aβ impairs mitochondrial function, which, in part, can subsequently trigger inflammatory cascades, creating a vicious cycle of neuronal damage. Estrogen receptors (ERs) are widely expressed throughout the brain, and the sex hormone 17β-estradiol (E2) exerts neuroprotection through both anti-inflammatory and mitochondrial mechanisms. While E2 exhibits neuroprotective properties, its mechanisms against Aβ toxicity remain incompletely understood. In this study, we investigated the neuroprotective effects of E2 against Aβ-induced mitochondrial dysfunction and neuroinflammation in primary cortical neurons, with a particular focus on the role of AMP-activated protein kinase (AMPK). We found that E2 treatment significantly increased phosphorylated AMPK and upregulated the expression of mitochondrial biogenesis regulator peroxisome proliferator-activated receptor gamma coactivator-1 α (PGC-1α), leading to improved mitochondrial respiration. In contrast, Aβ suppressed AMPK and PGC-1α signaling, impaired mitochondrial function, activated the pro-inflammatory nuclear factor kappa-light-chain enhancer of activated B cells (NF-κB), and reduced neuronal viability. E2 pretreatment also rescued Aβ-induced mitochondrial dysfunction, suppressed NF-κB activation, and, importantly, prevented the decline in neuronal viability. However, the pharmacological inhibition of AMPK using Compound C (CC) abolished these protective effects, resulting in mitochondrial collapse, elevated inflammation, and cell death, highlighting AMPK’s critical role in mediating E2’s actions. Interestingly, while NF-κB inhibition using BAY 11-7082 partially restored mitochondrial respiration, it failed to prevent Aβ-induced cytotoxicity, suggesting that E2’s full neuroprotective effects rely on broader AMPK-dependent mechanisms beyond NF-κB suppression alone. Together, these findings establish AMPK as a key mediator of E2’s protective effects against Aβ-driven mitochondrial dysfunction and neuroinflammation, providing new insights into estrogen-based therapeutic strategies for AD.

## 1. Introduction

Alzheimer’s disease (AD), the most prevalent neurodegenerative disorder, is marked by progressive synaptic loss, neuronal death, and cognitive impairments [1,2]. These changes lead to deficits in memory and behavior, ultimately interfering with daily functioning and quality of life [3]. In 2019, over 57 million people worldwide were affected by dementia, with AD accounting for approximately 70% of cases. This number is projected to rise to 152 million by 2050, posing a growing global public health crisis [4]. AD is characterized by two pathological hallmarks: extracellular amyloid-beta (Aβ) plaques and intracellular neurofibrillary tangles (NFTs) [5]. While the disease affects various brain regions, damage typically begins in the cortex and hippocampus, which are areas critical for higher cognitive functions, including memory formation [6,7]. In addition to Aβ and NFTs, recent research has identified mitochondrial dysfunction and chronic neuroinflammation as key contributors to AD progression [8,9,10]. These processes are interlinked, as mitochondrial impairment can initiate and/or exacerbate inflammatory signaling, and sustained neuroinflammation can, in turn, further disrupt mitochondrial function [11].

Some investigators contend that Aβ accumulation is a central pathological feature of AD and also contributes directly to mitochondrial dysfunction by disrupting oxidative phosphorylation (OXPHOS) and reducing adenosine triphosphate (ATP) production, ultimately causing synaptic failure and neuronal loss [12,13,14]. In parallel, Aβ also activates inflammatory signaling cascades within neurons [15] and involves the nuclear factor kappa-light-chain enhancer of activated B cell (NF-κB) pathway [16]. NF-κB, a key regulator of pro-inflammatory gene expression, is aberrantly activated in early AD, with elevated p65 levels observed in neurons and astrocytes near amyloid plaques [17,18]. Under normal conditions, NF-κB remains sequestered in the cytoplasm by its inhibitor, IκBα. Upon triggering by pathological stimuli such as Aβ, IκBα becomes phosphorylated and undergoes proteasomal degradation, allowing the release and nuclear translocation of NF-κB [19]. Once in the nucleus, NF-κB promotes the transcription of pro-inflammatory mediators, driving neuroinflammation and contributing to disease progression. Interestingly, NF-κB also influences mitochondrial function by regulating genes involved in mitochondrial biogenesis and metabolism [20]. Beyond its nuclear role, NF-κB subunits have been detected within mitochondria, suggesting a direct role in mitochondrial gene regulation [21,22,23,24]. Dysfunctional mitochondria release mitochondrial DNA (mtDNA) and other damage-associated molecular patterns (DAMPs), which trigger inflammatory responses [25]. This creates a vicious cycle where neuroinflammation and mitochondrial dysfunction reinforce each other, forming a self-perpetuating cycle of neurodegeneration [26].

Estrogen, a steroid hormone primarily known for its role in reproduction, also exerts significant effects on the central nervous system (CNS) [27]. Among the three main endogenous estrogens, 17β-estradiol (E2) is the most potent form [28]. Estrogen receptors (ERs) are widely distributed across brain regions, where they regulate neuronal growth, synaptic plasticity, and neuroinflammatory responses, thereby supporting neural health and cognitive function [29,30]. The widespread presence of ERs highlights the potential regulatory role of estrogen in maintaining brain health. E2 possesses significant neuroprotective properties [31] and promotes neuronal survival by upregulating neurotrophic factors, enhancing mitochondrial function, and attenuating oxidative stress [32,33,34,35]. E2 also regulates cellular energy homeostasis through the activation of AMP-activated protein kinase (AMPK) and its downstream target peroxisome proliferator-activated receptor gamma coactivator-1α (PGC-1α), both of which are essential for mitochondrial biogenesis [36,37]. Additionally, E2 exerts anti-inflammatory effects by suppressing NF-κB-mediated gene expression and modulating microglial activity, thereby limiting neuroinflammation [38,39]. Collectively, these multifaceted actions highlight the critical role of E2 in maintaining neuronal integrity in the context of neurodegenerative disorders.

AMPK plays a dual role in maintaining mitochondrial integrity and regulating inflammatory responses. By activating PGC-1α, AMPK promotes mitochondrial biogenesis and enhances OXPHOS, which are essential for neuronal energy homeostasis [40,41]. Additionally, AMPK exerts anti-inflammatory effects across various cell types, including human aortic smooth muscle cells (HASMCs) and macrophages [42,43,44]. Notably, E2 has been identified as an upstream activator of AMPK, thereby promoting metabolic efficiency and mitochondrial resilience [45,46]. In this study, we investigated whether E2 protects neurons from Aβ toxicity and explored the involvement of AMPK signaling and NF-κB-mediated inflammation in this process. Using primary cortical neurons exposed to Aβ, we assessed mitochondrial respiration, ATP production, inflammation, and neuronal viability. To delineate the contribution of each pathway, we employed the pharmacological inhibitors of AMPK and NF-κB. This approach enabled us to dissect the distinct and overlapping roles of AMPK and NF-κB signaling in mediating E2’s protective effects against Aβ toxicity. Overall, our findings suggest that E2 protects neurons from Aβ-induced mitochondrial dysfunction and inflammation through an AMPK-dependent mechanism. This not only advances our understanding of E2’s neuroprotective pathways but also identifies AMPK as a potential therapeutic target in AD. Targeting AMPK activation or modulating estrogen signaling may therefore offer a promising strategy for mitigating neurodegeneration in affected individuals.

## 2. Results

### 2.1. E2 Activates AMPK in a Dose- and Time-Dependent Manner and Enhances PGC-1α Levels

To assess whether E2 modulates energy metabolism in neurons, we examined its effect on AMPK activation. The treatment of primary cortical neurons with E2 at doses of 0.1 nM, 1 nM, and 10 nM for 1 h induced a significant, dose-dependent increase in AMPK phosphorylation (pAMPK) at Thr172 relative to the total AMPK (Figure 1A). Maximal activation was observed at 10 nM E2, which increased pAMPK levels by approximately 4-fold compared to untreated controls. A time-course study revealed that 10 nM E2 rapidly elevated pAMPK within 30 min (Figure 1B). Sustained AMPK activation persisted for up to 24 h (Figure 1C), indicating the long-term modulation of energy metabolism pathways. Given that AMPK is a key upstream regulator of PGC-1α [47], we assessed PGC-1α protein expression in response to E2. Acute exposure to 10 nM E2 for 30–60 min significantly increased PGC-1α levels (normalized to β-actin; Figure 1D). This elevation was maintained over prolonged treatments, with PGC-1α levels remaining elevated at 6 and 24 h (Figure 1E).

### 2.2. E2 Enhances Mitochondrial Respiration and Suppresses NF-κB Subunit Expression

To assess functional mitochondrial outcomes, we measured OCR in live cortical neurons using the Seahorse XF24 Analyzer. Treatment with 10 nM E2 for 6 and 24 h significantly increased both maximal respiration and spare respiratory capacity (Figure 1F,G), indicating enhanced mitochondrial efficiency and metabolic flexibility. In addition to its effects on energy metabolism, E2 also modulated inflammatory signaling pathways. Notably, E2 suppressed NF-κB activation, as evidenced by a time-dependent reduction in phosphorylated IκBα (pIκBα), a marker of NF-κB pathway activation (Figure 2). Acute E2 treatment for up to 60 min significantly reduced pIκBα levels (Figure 2A), with sustained suppression observed at 6 and 24 h (Figure 2B). Together, these findings suggest that E2 enhances mitochondrial function while also attenuating initial NF-κB-driven neuroinflammatory signaling.

### 2.3. Aβ Suppresses AMPK Activation, Reduces PGC-1α Levels, and Impairs Mitochondrial Function

To determine the pathological impact of Aβ on neuronal energy regulation, we examined AMPK signaling in primary cortical neurons. Exposure to Aβ induced a time-dependent suppression of AMPK phosphorylation in primary cortical neurons, with significant reductions observed at 3, 6, and 24 h (Figure 3A). Additionally, Aβ decreased PGC-1α levels by nearly 50% at 24 h (Figure 3B). The functional assessment of mitochondrial respiration using the Seahorse XF24 Analyzer revealed that Aβ impaired the OCR, significantly reducing basal respiration, maximal respiration, and ATP production (Figure 3C–F), consistent with mitochondrial dysfunction [14].

### 2.4. Aβ Activates NF-κB and Reduces Cell Viability

Aβ triggered NF-κB pathway activation, as demonstrated by a time-dependent reduction in pIκBα levels (which normally facilitates subsequent NF-κB nuclear translocation; Figure 3G). This proximal pro-inflammatory response coincided with a significant reduction in neuronal viability. Aβ exposure for 48 h reduced cell viability to about 60% of control levels (Figure 3H).

### 2.5. E2 Pretreatment Rescues Aβ-Induced Mitochondrial Dysfunction, Restores AMPK and PGC-1α Signaling, and Preserves Mitochondrial Electron Transport Chain Integrity

To investigate whether E2 pretreatment could counteract Aβ toxicity, primary cortical neurons were pretreated with 10 nM E2 for 48 h prior to 24 h Aβ exposure. Aβ significantly reduced AMPK phosphorylation, while E2 pretreatment maintained pAMPK levels (Figure 4A). Similarly, Aβ suppressed PGC-1α expression, but E2 pretreatment prevented the suppression of PGC-1α levels (Figure 4B), indicating the protection of pathways involved in cellular energy homeostasis and mitochondrial biogenesis. The functional assessment of mitochondrial respiration in live cells via the Seahorse XF24 Analyzer revealed that Aβ markedly impaired mitochondrial respiration, reducing basal respiration, maximal respiration, and spare respiratory capacity (Figure 4C–G). E2 pretreatment showed a trend toward reversing these impairments and improving all OCR parameters, although the changes did not reach statistical significance. Consistent with these findings, Aβ significantly reduced cellular ATP levels by approximately 50%, which were rescued by E2 pretreatment (Figure 4H). To further investigate the impact on mitochondrial structure and function, we analyzed the protein levels of ETC complex subunits. Aβ exposure to primary cortical neurons significantly decreased mitochondrial ETC complex subunit levels, including proteins of Complex I (NDUFB8), Complex II (SDHB), Complex IV (MTCO1), and Complex V (ATP5A) (Figure 5A–E). E2 pretreatment prevented this decline, restoring all four ETC complex subunit levels.

### 2.6. E2 Pretreatment Suppresses Aβ-Induced Neuroinflammation by Inhibiting NF-κB and Inflammasome Activity

To further investigate E2’s neuroprotective mechanisms, we examined its effects on Aβ-induced neuroinflammatory signaling. Aβ significantly triggered neuroinflammatory responses in primary cortical neurons, as shown by the increased phosphorylation of IκBα (Figure 6A). Aβ significantly enhanced the nuclear translocation of the NF-κB p65 subunit (Figure 6B) and its DNA binding activity (Figure 6C), indicating the sustained activation of the NF-κB pathway. While E2 alone significantly reduced phospho-IκBα levels, its effect on nuclear p65 levels showed a trend toward reduction but did not reach statistical significance (*p* = 0.06) (Figure 6A,B). E2 pretreatment significantly mitigated Aβ effects by reducing IκBα phosphorylation, suppressing the nuclear translocation of p65, and inhibiting NF-κB p65 DNA-binding activity compared to Aβ-treated neurons. Given that NF-κB activation can promote inflammasome assembly and caspase-1 activation [48], we next assessed caspase-1 activity. Aβ significantly increased caspase-1 activity (Figure 6D), a hallmark of NLRP3 inflammasome activation [49]. This increase was abolished by the selective caspase-1 inhibitor Ac-YVAD-CHO, confirming assay specificity. Importantly, E2 pretreatment significantly reduced Aβ-induced caspase-1 activity, indicating the suppression of inflammasome signaling. Consistent with this, Aβ elevated the levels of pro-inflammatory cytokines, including IL-1β, IL-6, TNFα, and IL-18, while reducing the anti-inflammatory cytokine IL-10 (Figure 6E). E2 pretreatment prevented the Aβ-induced increase in pro-inflammatory cytokines and restored the levels of anti-inflammatory cytokine IL-10 to those seen in control neurons. Collectively, these findings demonstrate that E2 attenuates Aβ-induced neuroinflammation by suppressing NF-κB activation, inhibiting inflammasome activity, and restoring a balanced cytokine profile.

### 2.7. E2 Pretreatment Rescues Aβ-Induced Cytotoxicity and Preserves Neuronal Viability During Prolonged Exposure

To evaluate the longer-term neuroprotective potential of E2, primary cortical neurons were pretreated with 10 nM E2 for 48 h before exposure to 10 μM Aβ for 48 h. Aβ significantly reduced neuronal viability, as measured by the MTT assay (Figure 7A), and increased cytotoxicity, as indicated by elevated LDH release (Figure 7B). E2 pretreatment maintained neuronal viability and significantly reduced LDH release compared to Aβ-treated neurons. Notably, E2 alone had no significant impact on viability or cytotoxicity, confirming its non-toxic effects under the experimental conditions.

### 2.8. AMPK Mediates E2’s Neuroprotective Effects Against Aβ-Induced Mitochondrial Dysfunction and Neurotoxicity

To confirm the role of AMPK in E2-mediated neuroprotection, we first optimized the dose of the AMPK inhibitor Compound C (CC). The pretreatment of primary cortical neurons with increasing doses of CC (1–5 μM) for 1 h prior to 10 nM E2 exposure revealed that 5 μM CC completely blocked E2-induced AMPK phosphorylation (Appendix A), consistent with previous studies demonstrating the efficacy of this concentration in neuronal systems [50]. In Aβ-treated neurons, AMPK phosphorylation and PGC-1α levels were significantly reduced (Figure 8A,B). E2 pretreatment maintained both pAMPK and PGC-1α levels; however, this rescue was completely abolished by CC, confirming the requirement of AMPK for E2-mediated PGC-1α upregulation (Figure 8A,B). Functionally, Aβ significantly impaired mitochondrial respiration, as indicated by reductions in basal respiration, maximal respiration, and ATP production, which was rescued by E2 pretreatment (Figure 8C–F). These protective effects of E2 were fully reversed by CC, with OCR parameters reduced to levels comparable to the Aβ treatment alone (Figure 8C–F). Consistent with mitochondrial dysfunction, Aβ significantly reduced neuronal viability and increased cytotoxicity in our cultures, which was prevented by E2 pretreatment (Figure 8G,H). However, this rescue was abolished when AMPK was inhibited using CC, which completely blocked E2’s protective effect, reducing viability to the levels observed with Aβ alone. Collectively, these findings establish that AMPK activation is essential for E2’s ability to counteract Aβ-induced mitochondrial dysfunction and neurotoxicity.

### 2.9. E2 Acts via AMPK to Suppress Aβ-Induced NF-κB Subunit Expression

We next assessed whether E2’s anti-inflammatory effects were AMPK-dependent. Aβ exposure significantly increased phosphorylated IκBα levels and promoted the nuclear translocation of the NF-κB p65 subunit, indicating the robust activation of the NF-κB pathway (Figure 9A,B). E2 pretreatment attenuated these effects by reducing both pIκBα levels and nuclear p65 translocation compared to Aβ-treated neurons. However, the AMPK inhibitor CC abolished E2’s suppressive effect on NF-κB activation, restoring pIκBα and nuclear p65 levels to those observed with Aβ treatment alone (Figure 9A,B). These findings confirm that AMPK activation is essential for the E2-mediated suppression of NF-κB signaling, placing AMPK upstream of NF-κB in E2’s anti-inflammatory mechanism.

### 2.10. NF-κB Inhibition Partially Rescues Mitochondrial Dysfunction but Fails to Prevent Aβ-Induced Cytotoxicity

Aβ exposure significantly impaired mitochondrial respiration, as revealed by significant reductions in basal respiration, maximal respiration, ATP production, and spare respiratory capacity (Figure 10A–E). Consistent with previous findings, E2 pretreatment significantly rescued these parameters. The NF-κB inhibitor BAY 11-7082, at an optimized dose determined in our culture system (Appendix A), also rescued mitochondrial function similarly to E2 pretreatment and restored maximal respiration, ATP production, and spare respiratory capacity (Figure 10C–E), although it showed only a non-significant trend towards improving basal respiration (Figure 10B). In contrast, BAY 11-7082 failed to protect against Aβ-induced cytotoxicity. Aβ significantly reduced neuronal viability (Figure 10F) and increased cytotoxicity (Figure 10G) in our cultures, and while the E2 pretreatment prevented this, BAY 11-7082 had no protective effect.

## 3. Discussion

In this study, we investigated the neuroprotective effects of E2 in primary cortical neurons exposed to Aβ, with a focus on AMPK signaling and NF-κB-mediated inflammation. AD is marked by the progressive accumulation of Aβ in the cortex [6,51], and to recapitulate this pathological hallmark in vitro, we treated cortical neurons derived from mouse embryos with Aβ, as established in previous studies [52,53,54]. Although E2 has been previously shown to confer neuroprotection, our findings offer novel insights into the mechanisms underlying this effect in the context of Aβ-induced toxicity. We demonstrated that E2 robustly activates AMPK signaling and its downstream effector PGC-1α (Figure 1A–E), promoting mitochondrial respiration (Figure 1F–H) and enhancing ETC complex protein expression (Figure 5A–E). These results align with previous findings highlighting AMPK’s role in supporting mitochondrial function via PGC-1α [40,41,55].

E2 also reduced NF-κB activity by attenuating IκBα phosphorylation (Figure 2A,B), consistent with its established anti-inflammatory properties [38,56]. Aβ exposure led to a significant reduction in pAMPK and PGC-1α levels (Figure 3A,B), impaired mitochondrial respiration (Figure 3C–F), and increased NF-κB activation via the enhanced phosphorylation and degradation of its inhibitor IκBα (Figure 3G), culminating in pronounced neurotoxicity (Figure 3H). E2 pretreatment effectively reversed these pathological changes: It suppressed NF-κB activation (Figure 6A–C), attenuated caspase-1 activity and pro-inflammatory cytokine expression (Figure 6D,E), restored AMPK and PGC-1α levels (Figure 4A,B), rescued mitochondrial deficits (Figure 4C–H), and improved neuronal viability (Figure 7A,B). Crucially, the pharmacological inhibition of AMPK abolished E2-mediated mitochondrial and neuroprotective effects (Figure 8A–G and Figure 9A,B), establishing AMPK as a pivotal mediator of E2’s actions. Interestingly, the inhibition of NF-κB partially rescued mitochondrial respiration but did not prevent Aβ-induced neuronal death (Figure 10A–F), suggesting that E2’s effects involve AMPK-dependent pathways that extend beyond NF-κB suppression. Together, these results highlight AMPK as a central player in the neuroprotective actions of E2 against Aβ-induced neurotoxicity.

E2 exerts its neuroprotective effects through both genomic and non-genomic signaling pathways [57], mediated by ERs that are widely expressed in the brain [58]. The genomic actions of ERα and ERβ control the transcription of neuroprotective factors, including but not limited to brain-derived neurotrophic factor (BDNF), PGC-1α, and even anti-apoptotic proteins [59,60,61]. Non-genomic mechanisms involve the rapid activation of membrane-bound ERs or GPER, triggering downstream protein kinase cascades, including AMPK activation, to modulate gene expression [36,62]. All three ERs, ERα, ERβ, and GPER, contribute to the neuroprotective effects of E2 via both pathways [63,64,65]. For instance, ERα has been implicated in mediating E2’s anti-inflammatory effects in the brain [66] and in providing protection against strokes in both male and female mice [65], while ERβ supports learning and memory in the postmenopausal models of AD by maintaining mitochondrial homeostasis through NRF1/PGC-1α signaling [61]. GPER and ERα both enhance mitochondrial function [67], and studies in ERα and ERβ knockout mice further confirm their roles in modulating inflammatory cytokine production [68]. These findings collectively support the idea that E2’s neuroprotective actions are mediated by a complex interplay of receptor subtypes. Notably, ERs localize in mitochondria [69], where they may directly regulate mitochondrial function and contribute to estrogen’s protective effects. In our study, it is likely that both pathways mediate the observed benefits of E2. However, the complete loss of E2’s protective effects upon AMPK inhibition (Figure 8 and Figure 9) underscores the critical role of membrane-initiated non-genomic signaling in coordinating mitochondrial function and inflammation. This aligns with reports that E2 rapidly activates AMPK through non-genomic mechanisms [70] and with prior reviews emphasizing the role of non-genomic signaling in mediating E2’s neuroprotective effects [71].

AMPK regulates PGC-1α through the following: direct phosphorylation to enhance its transcriptional activity [47] and indirect activation via Sirtuin 1 (SIRT1)-mediated deacetylation [41]. This cascade was evident in our study, where E2 enhanced mitochondrial respiration and function (Figure 1). AMPK inhibition abolished E2’s ability to restore PGC-1α levels (Figure 8B), indicating that AMPK functions upstream of PGC-1α in primary cortical neurons and is essential for E2-driven mitochondrial biogenesis. Concurrently, E2 suppressed NF-κB activation, as evidenced by reduced IκBα phosphorylation (Figure 2), consistent with its anti-inflammatory effects [56,72]. Together, these findings highlight E2’s ability to coordinate mitochondrial and inflammatory regulation likely through AMPK-PGC-1α signaling. Aβ, a key pathogenic factor in AD, is linked to mitochondrial dysfunction, neuroinflammation, and neuronal death. Its toxicity varies with its aggregation state [73]. Monomers are generally non-toxic, but oligomers and fibrils disrupt synaptic function and exacerbate neuroinflammation by activating pro-inflammatory pathways like NF-κB [74], leading to pro-inflammatory cytokine release and chronic inflammation. Aβ also impairs mitochondrial function, as observed in our study where Aβ resulted in decreased pAMPK and PGC-1α levels, impaired mitochondrial respiration (Figure 3A–F), and enhanced NF-κB activation (Figure 3G), resulting in neuronal death (Figure 3H). These findings align with previous studies demonstrating Aβ-induced mitochondrial dysfunction [12] and cell death [75,76], underscoring the pathogenic role of Aβ in AD.

Consistent with our hypothesis, E2 pretreatment effectively counteracted Aβ-induced mitochondrial and inflammatory deficits, demonstrating its neuroprotective potential. Specifically, E2 restored pAMPK and PGC-1α levels (Figure 4A,B), improved mitochondrial respiration (Figure 4C–G), rescued ATP production (Figure 4H), and enhanced ETC complex expression (Figure 5A–E). Mechanistically, the E2-mediated activation of AMPK and upregulation of PGC-1α likely promote mitochondrial biogenesis through the transcriptional induction of key downstream factors, such as nuclear respiratory factor 1 (NRF1), nuclear respiratory factor 2 (NRF2), and mitochondrial transcription factor A (TFAM) [77,78]. Additionally, E2 enhances mitochondrial antioxidant capacity by upregulating superoxide dismutase 2 (SOD2) [50] and stabilizes calcium homeostasis, countering the Aβ-induced calcium overload that exacerbates mitochondrial dysfunction [79,80,81].

In our study, Aβ exposure increased the nuclear translocation of the NF-κB subunit p65 (Figure 3G) and its DNA-binding activity (Figure 6C), reflecting heightened inflammatory signaling. E2 pretreatment suppressed Aβ-enhanced NF-κB responses (Figure 6A–C), supporting prior studies demonstrating E2’s inhibition of NF-κB in macrophages and rat aortic smooth muscle cells (RASMCs) [38,82]. This suggests that E2’s rescue of mitochondrial dysfunction in primary neurons involves NF-κB inhibition and the subsequent attenuation of inflammatory cascades. Aβ further triggered caspase-1 activity and elevated pro-inflammatory cytokines (Figure 6D,E), processes associated with NLRP3 inflammasome activation and IL-1β/IL-18 maturation [83,84]. E2 mitigated these effects, reducing caspase-1 activity and pro-inflammatory cytokine levels, thereby attenuating neuroinflammatory markers. E2 pretreatment also improved the Aβ-mediated decline in cell viability (Figure 7A,B), likely through its dual actions on mitochondrial function and inflammation. This supports earlier findings showing that E2 protects against Aβ toxicity. For instance, in rat hippocampal neurons, E2 prevented Aβ-induced apoptosis by upregulating the anti-apoptotic Bcl-2, preventing Bax translocation to mitochondria, and maintaining calcium homeostasis [80]. Similarly, in the SN56 cholinergic cells, E2 prevented cell death induced by Aβ [85]. Building on this, our findings suggest that rescuing mitochondrial respiration and ETC complex proteins (Figure 4 and Figure 5) would enhance energy metabolism, while suppressing NF-κB and caspase-1 (Figure 6) would reduce inflammatory stress, collectively contributing to improved neuronal survival.

Although NF-κB plays a pivotal role in Aβ-induced neuroinflammation, it is not the sole contributor. Other pathways, including the Janus kinase/signal transducer and activator of transcription (JAK/STAT), mitogen-activated protein kinases (MAPKs), and the NLRP3 inflammasome, also mediate Aβ toxicity [86,87,88,89,90,91]. Importantly, E2 regulates these pathways [92,93], indicating a broader anti-inflammatory role beyond NF-κB suppression. This could explain why E2, unlike NF-κB inhibition, rescued both mitochondrial function and neuronal cell death. To elucidate NF-κB’s role in Aβ-induced neuronal damage, we used the selective NF-κB inhibitor BAY11-7082 [94], which blocks IκBα phosphorylation and degradation, preventing NF-κB nuclear translocation [95]. NF-κB inhibition partially restored mitochondrial function to levels comparable to the E2 treatment (Figure 10A–E). This aligns with previous findings implicating NF-κB in mitochondrial dysfunction across various models [96]. NF-κB mediates Aβ-induced mitochondrial impairment by reducing mitochondrial function and cytochrome c oxidase III (COX III) activity [97]. NF-κB can translocate into mitochondria and bind mitochondrial DNA (mtDNA), repressing genes such as cytochrome b and COX III [24,98,99]. Despite restoring mitochondrial function, BAY11-7082 failed to prevent Aβ-induced neuronal cell death (Figure 10F,G), suggesting that NF-κB inhibition alone is insufficient for full neuroprotection. This indicates that mitochondrial dysfunction is only one aspect of Aβ toxicity. Other mechanisms, such as calcium dysregulation [100], oxidative stress [101], and apoptotic signaling pathways [102], may also contribute to neuronal death.

The activation of AMPK signaling downregulates NF-κB activity through downstream targets such as SIRT1 and PGC-1α, thereby reducing inflammation [103,104,105]. We found that Aβ exposure lowered the levels of pAMPK and PGC-1α, but the E2 pretreatment reversed these effects. However, when AMPK was inhibited with CC prior to the E2 treatment, it abolished E2’s protective effects, preventing the restoration of pAMPK and PGC-1α (Figure 8A,B), impairing mitochondrial respiration (Figure 8C–F), and reducing neuronal survival (Figure 8G,H). Similarly, CC blocked E2’s suppression of NF-κB activation (Figure 9A,B). Collectively, these findings position AMPK as a central upstream regulator of PGC-1α-dependent mitochondrial function and NF-κB-mediated inflammatory signaling in primary cortical neurons, highlighting its pivotal role in mediating the neuroprotective effects of E2. AMPK is therefore a likely key mediator of E2’s neuroprotective effects against Aβ-induced toxicity, coordinating mitochondrial function and anti-inflammatory responses.

Despite the valuable insights gained from our study, several limitations must be acknowledged. Our experiments were conducted in vitro, which, although enabling detailed mechanistic analysis, lacks the complexity of the intact brain. While animal and cell culture models of Aβ toxicity have contributed substantially to AD research [106,107], they often focus primarily on amyloid accumulation and fail to replicate the full spectrum of human pathology, particularly NFTs. Tau pathology plays a central role in AD and is strongly associated with neuronal loss and cognitive decline [108,109], yet it was not examined in this study. While our results support Aβ-induced mitochondrial and neuronal dysfunction, the specific mitochondrial sites or mechanisms involved were not directly assessed. Prior studies suggest that Aβ may enter mitochondria via the TOM/TIM complex [110], but this interaction remains unexplored in our model. Moreover, Aβ plaque burden in the human brain does not consistently correlate with cognitive impairment [111], highlighting the importance of additional factors such as tau in disease progression. Future research should address these gaps using in vivo models and tau-related endpoints to provide a more comprehensive understanding of AD pathogenesis.

Clinically, these findings highlight the therapeutic potential of targeting AMPK and estrogen signaling pathways in AD. While clinical trials on estrogen replacement therapy have produced mixed results [112,113,114,115], a combinatorial approach integrating AMPK activators with estrogen-based treatments may yield improved outcomes. Importantly, the higher risk of AD in postmenopausal women [116,117] underscores the relevance of estrogen’s neuroprotective actions in this population. In vivo studies using postmenopausal models will be essential to validate these findings in a physiologically relevant context. Such models could also help delineate the interplay between systemic estrogen loss and AMPK signaling in AD pathogenesis. Investigating the optimal timing, formulation, and delivery of estrogen therapies, as well as evaluating how AMPK-targeted strategies might benefit both sexes, will be important for developing effective interventions to prevent disease progression.

## 4. Materials and Methods

### 4.1. Cell Culture

Primary neuron cultures were prepared from the cortices of embryonic day 18.5 (E18.5) C57BL/6N mice using methods previously described [118]. All animal procedures were conducted in accordance with the guidelines set by the University of Manitoba’s Animal Care Committee and the Canadian Committee on Animal Care (CCAC). Cells were seeded onto poly-D-lysine-coated cell culture plates at specific densities: 500,000 cells per well for 12-well plates and 50,000 cells per well for 96-well plates. The culture medium used was Neurobasal-A Medium, with no D-glucose and no sodium pyruvate (21103049, Thermo Fisher Scientific, Waltham, MA, USA), and it was supplemented with 3 mM D-glucose (G7528, MilliporeSigma Canada Ltd., Oakville, ON, Canada), 0.22 mM sodium pyruvate (11360070, Thermo Fisher Scientific, Waltham, MA, USA), 1 × B-27 (A3582801, Thermo Fisher Scientific, Waltham, MA, USA), 100 µg/mL penicillin-streptomycin (SV30010, Thermo Fisher Scientific, Waltham, MA, USA), and 5 mM HEPES (SH3023701, Thermo Fisher Scientific, Waltham, MA, USA) (NB media). Five percent heat-inactivated charcoal-stripped fetal bovine serum (FBS) and 1 × Glutamax (35050079, Thermo Fisher Scientific, Waltham, MA, USA) were added during initial seeding. FBS was removed from the cultures the following day. To prevent glial cell proliferation, cytosine arabinofuranoside (2 μM) (C1768, MilliporeSigma Canada Ltd., Oakville, ON, Canada) was added to the cultures on day 2. The culture media were replaced on day 3 with fresh NB media supplemented with 1 × Glutamax. Cultures were maintained in a humidified incubator at 37 °C with 5% CO_2_. The media were partially changed every 48 h, and cultures were used for experiments on day 7. For pharmacological interventions, the following drugs and inhibitors were used: 10 nM 17 beta-estradiol (E2) (E2758, MilliporeSigma Canada Ltd., Oakville, ON, Canada); 10 μM Aβ (A-1163-2, rPeptide, Watkinsville, GA, USA); 5 μM compound C (CC) (171261, MilliporeSigma Canada Ltd., Oakville, ON, Canada) to inhibit AMPK; and 3 μM BAY 11-7082 (196871, MilliporeSigma Canada Ltd., Oakville, ON, Canada) to inhibit the phosphorylation of IκBα (thus blocking NF-κB activation).

### 4.2. Amyloid Beta Preparation

Monomeric Aβ_1–42_ peptide was obtained from rPeptide (A-1163, rPeptide, Watkinsville, GA, USA; sequence: DAEFRHDSGYEVHHQKLVFFAEDVGSN KGAIIGLMVGGVVIA), expressed recombinantly in *E. coli*, and purified to >97% by mass spectrometry to ensure high purity and consistency. The peptide was supplied as a clear dry film, pretreated with hexafluoroisopropanol (HFIP), which disrupts pre-existing aggregates and yields a highly monomeric starting material. The lyophilized peptide film was resuspended in dimethyl sulfoxide (DMSO) (276855, MilliporeSigma Canada Ltd., Oakville, ON, Canada) to a final concentration of 10 mM and sonicated in a water bath for 10 min to ensure complete dissolution. The resulting stock solution was aliquoted and stored at –80 °C. For oligomer formation, the 10 mM stock was diluted to 10µM in cell culture media and incubated at 4 °C for 24 h, following a published protocol with minor modifications [119]. This protocol is widely used to generate Aβ oligomers, which are considered the most neurotoxic species relevant to AD pathology. The Aβ oligomer solution was used to treat cells, ensuring that the final DMSO concentration was less than 0.1%.

### 4.3. Quantitative Western Blotting

To analyze protein expression levels, cell lysates were harvested and homogenized in an ice-cold RIPA buffer (ab156034, Abcam, Waltham, MA, USA) containing 1% nonylphenol, 0.8% sodium chloride, 0.6% Tris (pH 7.5), 0.5% sodium deoxycholate, 0.1% sodium pyrophosphate decahydrate, 0.11% sodium lauryl sulfate, 0.03% EGTA, 0.03% EDTA, 0.03% disodium dihydroxypropanyl phosphate, 0.01% sodium orthovanadate, and a protease and phosphatase inhibitor cocktail (78440, Thermo Fisher Scientific, Waltham, MA, USA). Nuclear proteins were extracted using a nuclear extraction kit (78833, Thermo Fisher Scientific, Waltham, MA, USA) according to the manufacturer’s instructions. A detergent-compatible (DC) colorimetric protein assay kit (Bio-Rad, Hercules, CA, USA) was used to determine protein concentrations. Proteins were resolved and separated by 10% sodium dodecyl sulfate–polyacrylamide gel electrophoresis (SDS-PAGE). Following electrophoresis, the proteins were transferred onto a nitrocellulose membrane (1620112, Bio-Rad, Hercules, CA, USA) using the Trans-Blot Turbo Transfer System (Bio-Rad, Hercules, CA, USA). Membranes were immunoblotted with specific primary antibodies overnight at 4 °C. We also used the following primary antibodies: phosphorylated AMPK (pAMPK) Thr172 (1:1000, Cell Signaling, Danvers, MA, USA), total AMPK (T-AMPK) (1:5000, Abcam, Waltham, MA, USA), Total OxPhos (1:1000, Thermo Fisher Scientific, Waltham, MA, USA; antibody cocktail against complexes I to V), β-actin (1:1000, Cell Signaling, Danvers, MA, USA), PGC-1α (1:1000, MilliporeSigma Canada Ltd., Oakville, ON, Canada), Phospho-IKβα (1:1000, Cell Signaling, Danvers, MA, USA), NF-κB p65 (1:1000, Santa Cruz Biotechnology Inc., Dallas, TX, USA), and Lamin A/C (1:1000, Santa Cruz Biotechnology Inc., Dallas, TX, USA). After primary antibody incubation, membranes were incubated with HRP-conjugated secondary antibodies, goat anti-rabbit IgG (H + L) or goat anti-mouse IgG (H + L) at a 1:5000 dilution (Jackson ImmunoResearch Laboratories, West Grove, PA, USA), for 1 h at room temperature. Total protein was visualized using the chemiluminescent imaging of the blot following gel activation (TGX Stain-Free FastCast Acrylamide Solutions, Bio-Rad, Hercules, CA, USA) in addition to β-actin for target protein normalization. Blots were developed using Clarity Western ECL Substrate (Bio-Rad, Hercules, CA, USA) or SignalFire ECL Reagent (Cell Signaling, Danvers, MA, USA) and imaged using a ChemiDoc MP imaging system (Bio-Rad, Hercules, CA, USA).

### 4.4. Seahorse Assay to Measure Mitochondrial Respiration

Mitochondrial respiration in primary embryonic cortical neurons was assessed using the Seahorse XF24 Analyzer (Agilent Technologies Inc., Santa Clara, CA, USA). Neurons were cultured on poly-D-lysine-coated Seahorse XF24 plates at a density of 50,000 cells per well for a total of 7 days before treatments were added and the experiment was conducted. One hour prior to the assay, the culture medium was replaced with unbuffered DMEM (Dulbecco’s modified Eagle’s medium, pH 7.4) supplemented with 0.22 mM sodium pyruvate and 3 mM D-glucose. The Seahorse assay was used to measure the basal mitochondrial oxygen consumption rate (OCR), maximal respiration, and spare respiratory capacity. Three drug cocktails containing mitochondrial complex inhibitors were sequentially injected through the ports of the Seahorse FluxPak cartridges: oligomycin (1 µM), carbonyl cyanide *p*-trifluoro methoxyphenylhydrazone (FCCP) (2 µM), and rotenone/antimycin A (1 µM). Oligomycin irreversibly inhibits ATP synthase, FCCP acts as an uncoupler, and rotenone and antimycin A inhibit Complex I and III of the electron transport chain (ETC), respectively. All four drugs were obtained from MilliporeSigma, Canada, Oakville, ON, Canada. Following OCR measurements, cell lysates were harvested individually from each well in a RIPA buffer, and a DC protein assay was performed. OCR measurements were normalized to the amount of protein in each well and reported as pmol/min/mg protein.

### 4.5. MTT Assay

Cell viability was assessed using the 3-(4,5-dimethylthiazol-2-yl)-2,5-diphenyl tetrazolium bromide (MTT) assay (ab211091, Abcam, Waltham, MA, USA) according to the manufacturer’s protocol. This colorimetric assay relies on the reduction of MTT to formazan by metabolically active cells, providing a quantitative measure of cell viability based on metabolic activity. Primary cortical neurons were seeded onto poly-D-lysine-coated 96-well plates and cultured under conditions previously described. Cells were pretreated with or without 10 nM E2 for 48 h, followed by treatment with 10 μM Aβ for an additional 48 h. After treatment, the media was discarded, and 50 μL of serum-free media, along with 50 μL of the MTT reagent, was added to each well. The plate was incubated at 37 °C for 3 h. Following incubation, 150 μL of MTT solvent was added to each well. The plate was wrapped in foil to protect from light and shaken for 30 min to ensure the complete solubilization of the formazan crystals. Absorbance was measured at 590 nm using a BioTek Synergy Neo2 (Agilent Technologies Inc., Santa Clara, CA, USA) plate reader to determine cell viability.

### 4.6. LDH Assay

Cytotoxicity was evaluated using the LDH Cytotoxicity Assay Kit (C20300, Thermo Fisher Scientific, Waltham, MA, USA), following the manufacturer’s instructions. This colorimetric assay relies on the conversion of lactate to pyruvate by lactate dehydrogenase (LDH), which is released into the culture medium upon cell membrane damage, providing a quantitative measure of cytotoxicity based on the extent of cellular damage. Primary cortical neurons were seeded onto 96-well plates and treated as previously described. After treatment, 50 μL of media from each well was transferred to a new 96-well plate, and 50 μL of freshly prepared reaction mixture was added. The plate was gently mixed while avoiding bubble formation and incubated at room temperature for 30 min in the dark. Following incubation, 50 μL of stop solution was added to each well and mixed by gentle tapping. Absorbance was measured at 490 nm and 680 nm using the BioTek Synergy Neo2 microplate reader to determine LDH activity, which correlates with the level of cellular damage.

### 4.7. NF-κB p65 Activation Assay

NF-κB activation was assessed using the NF-κB p65 Transcription Factor Assay Kit (ab133112, Abcam, Waltham, MA, USA), an enzyme-linked immunosorbent assay (ELISA)-based method that quantifies the binding of the NF-κB p65 subunit to a specific double-stranded DNA (dsDNA) sequence containing the NF-κB response element immobilized on a 96-well plate. Nuclear extracts were prepared from control and treated primary cortical neurons using the Nuclear Extraction Kit (78833, Thermo Fisher Scientific, Waltham, MA, USA) according to the manufacturer’s instructions. Extracts were added to the wells and incubated overnight at 4 °C to allow the specific binding of NF-κB p65 to the immobilized DNA. After washing, a primary antibody specific to NF-κB p65 (provided in the kit) was added and incubated for 1 h at room temperature on an orbital shaker. Following additional washes, an HRP-conjugated secondary antibody was added and incubated under the same conditions. After final washes, 100 μL of developing solution was added to each well and incubated for 30 min at room temperature with shaking. The reaction was stopped with 100 μL of stop solution, and absorbance was measured at 450 nm using a microplate reader to quantify NF-κB p65 activation. Data normalization was performed using total protein concentrations of the nuclear extracts, measured by a DC protein assay.

### 4.8. Measurement of Caspase-1 Activity

Caspase-1 is activated downstream of the NLRP3 inflammasome and plays a central role in the maturation of pro-inflammatory cytokines such as IL-1β. Caspase-1 activity, a hallmark of inflammasome activation, was quantified using the Caspase-Glo-1 Inflammasome Assay Kit (G9951, Promega, Madison, WI, USA) according to the manufacturer’s protocol. Neurons were seeded on 96-well plates and treated as previously described. Following treatment, the plates were removed from the incubator and equilibrated to room temperature for 10 min. Next, Caspase-Glo-1 Reagent or Caspase-Glo-1 YVAD-CHO Reagent was added to each well. The Caspase-Glo-1 Reagent contains MG-132 to inhibit proteasome-related nonspecific activity, while the Caspase-Glo-1 YVAD-CHO Reagent includes both MG-132 and the selective caspase-1 inhibitor Ac-YVAD-CHO to confirm the caspase-1 specificity of the observed luminescent signal. Plates were gently mixed on a plate shaker at 450 rpm for 5 min and then incubated at room temperature for 1 h and protected from light by wrapping in foil. Luminescence was measured using a microplate reader, and signal intensity was used to assess caspase-1 activity.

### 4.9. ATP Measurement

Cellular ATP levels were quantified using the luminescent ATP Determination Kit (A22066, Thermo Fisher Scientific, Waltham, MA, USA), which utilizes firefly luciferase to catalyze the reaction of D-luciferin with ATP, producing light. The luminescence generated is directly proportional to ATP concentrations, enabling the sensitive detection of cellular energy levels. Cells were cultured and treated as described. After treatment, cultures were washed with ice-cold PBS and lysed to release intracellular ATP. The lysates were centrifuged at 14,000 rpm for 10 min, and the supernatants were collected for ATP quantification according to the manufacturer’s guidelines. Appropriate background controls were included to subtract background luminescence. Samples were read at 560 nm using a microplate reader. ATP levels were normalized to total protein levels in each sample.

### 4.10. ELISA

Cytokine levels in cell culture supernatants were measured using commercially available ELISA kits for mouse IL-10 (ab100697), TNFα (ab208348), IL-6 (ab222503) (all from Abcam, Waltham, MA, USA), IL-18 (BMS618-3, Thermo Fisher Scientific, Waltham, MA, USA), and IL-1β (BMS6002, Thermo Fisher Scientific, Waltham, MA, USA) following the manufacturers’ instructions. Interleukin concentrations were calculated based on standard curves generated for each respective cytokine and reported as fold change relative to the untreated control group.

### 4.11. Statistical Analysis

Data are presented as mean ± standard error of the mean (SEM) and analyzed using one-way ANOVA followed by Dunnett’s or Tukey’s post hoc tests, as indicated. For comparisons between multiple treatment groups, one-way ANOVA followed by Tukey’s post hoc test was used. Statistical analyses were performed using GraphPad Prism (version 8.0) software. A *p* value of less than 0.05 (<0.05) was considered statistically significant in all experiments.

## Figures and Tables

**Figure 1 ijms-26-06203-f001:**
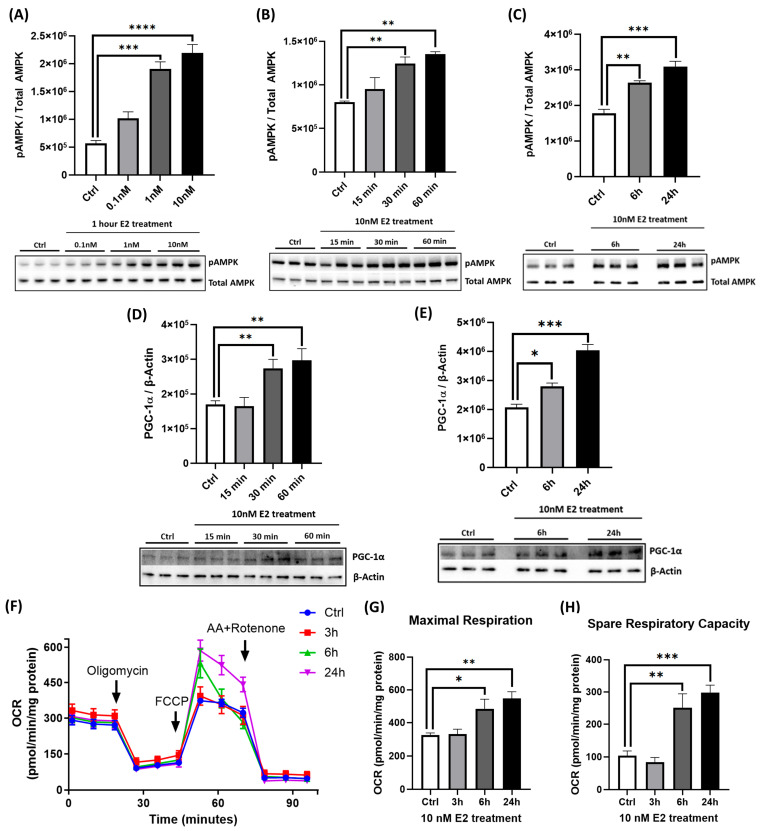
E2 activates AMPK, increases PGC-1α levels, and enhances mitochondrial function in primary cortical neurons. Primary cortical neurons were treated with or without E2 at various doses (0.1–10 nM) and for different durations (15 min to 24 h) as indicated. E2 was dissolved in DMSO (vehicle, final concentration <0.1%). Cell lysates were analyzed by Western blotting for (**A**–**C**) pAMPK (normalized to total AMPK) and (**D**,**E**) PGC-1α (normalized to β-actin). (**F**–**H**) Mitochondrial oxygen consumption rate (OCR), maximal respiration, and spare respiratory capacity were measured via a Seahorse XF24 Analyzer after 3 h, 6 h, and 24 h E2 treatment, and OCR (O_2_ in pmol/min) was normalized to protein concentration per well. Data are expressed as mean ± SEM, with n = 3 replicate cultures for Western blot analyses (**A**–**E**) and n = 5 replicate cultures for Seahorse assays (**F**–**H**). Statistical significance was determined by one-way ANOVA with Dunnett’s post hoc test; * *p* < 0.05, ** *p* < 0.01, *** *p* < 0.001, and **** *p* < 0.0001.

**Figure 2 ijms-26-06203-f002:**
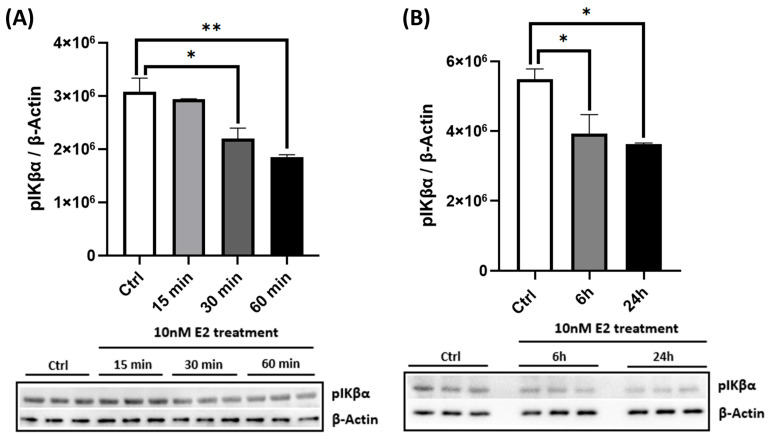
E2 suppresses NF-κB signaling in primary cortical neurons. Primary cortical neurons were treated with 10 nM E2 for various durations (15 min to 24 h). Cell lysates were analyzed by Western blotting for (**A**) acute (15–60 min) and (**B**) prolonged (6–24 h) changes in phosphorylated IκBα (pIκBα) levels, normalized to β-actin. Data are expressed as mean ± SEM, with n = 3 replicate cultures. Statistical significance was determined by one-way ANOVA with Dunnett’s post hoc test; * *p* < 0.05 and ** *p* < 0.01.

**Figure 3 ijms-26-06203-f003:**
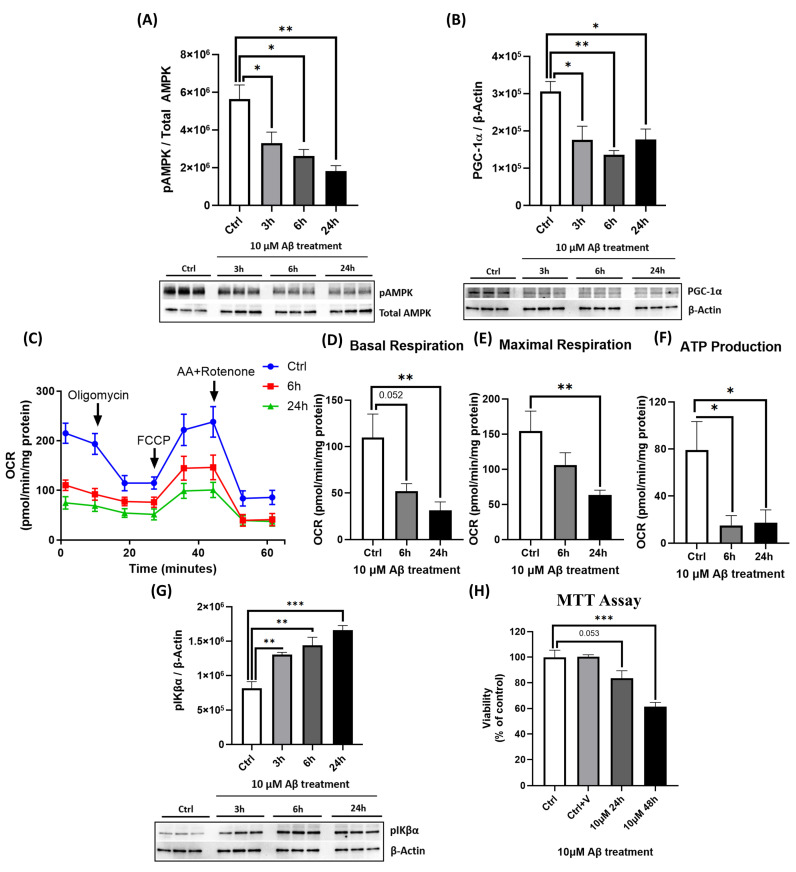
Aβ suppresses AMPK activation, decreases PGC-1α levels, impairs mitochondrial function, activates NF-κB, and reduces cell viability in primary cortical neurons. Primary cortical neurons were treated with 10 μM Aβ for 3, 6, or 24 h as indicated. Cell lysates were analyzed by Western blotting for (**A**) pAMPK (normalized to total AMPK), (**B**) PGC-1α (normalized to β-actin), and (**G**) pIκBα (normalized to β-actin). (**C**–**F**) OCR, including basal respiration, maximal respiration, and ATP production, was measured using the Seahorse XF24 Analyzer after 6 and 24 h of Aβ treatment (OCR was normalized to protein concentration per well). (**H**) Cell viability was assessed by an MTT assay after 24 or 48 h of Aβ treatment, with data shown as a percentage of the control group. Data are expressed as mean ± SEM, with n = 3 replicate cultures for Western blot analyses (**A**,**B,G**), n = 6–7 replicate cultures for Seahorse assays (**C**–**F**), and n = 4 replicate cultures for MTT assay (**H**). Statistical significance was determined by one-way ANOVA with Dunnett’s post hoc test; * *p* < 0.05, ** *p* < 0.01, and *** *p* < 0.001.

**Figure 4 ijms-26-06203-f004:**
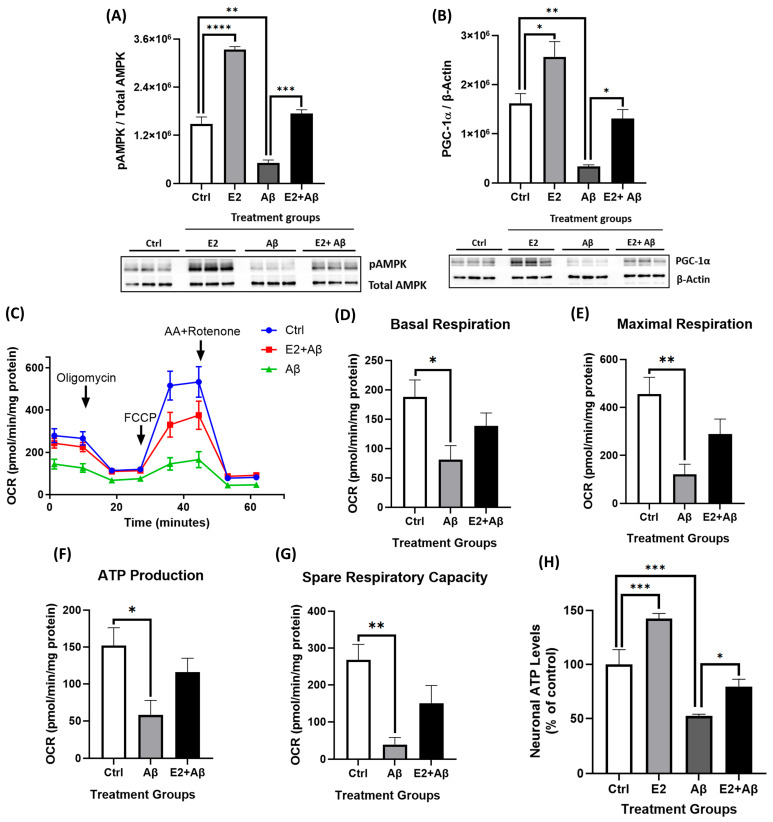
E2 pretreatment rescues Aβ-induced mitochondrial dysfunction, restores AMPK and PGC-1α levels, and enhances mitochondrial bioenergetics. Primary cortical neurons were pretreated with 10 nM E2 for 48 h before exposure to 10 μM Aβ for 24 h. (**A**) pAMPK levels (normalized to total AMPK) and (**B**) PGC-1α levels (normalized to β-actin) were analyzed by Western blotting. (**C**–**G**) OCR, including basal respiration, maximal respiration, ATP production, and spare respiratory capacity, was measured using the Seahorse XF24 Analyzer and normalized to protein concentration per well. (**H**) Cellular ATP levels (normalized to total protein) were measured using a luminescent ATP determination kit. Data are expressed as mean ± SEM, with n = 3 replicate cultures for Western blot analyses (**A**,**B**) and ATP assays (**H**) and n = 6–7 replicate cultures for Seahorse assays (**C**–**G**). Statistical significance was determined by one-way ANOVA with Tukey’s post hoc test; * *p* < 0.05, ** *p* < 0.01, *** *p* < 0.001, and **** *p* < 0.0001.

**Figure 5 ijms-26-06203-f005:**
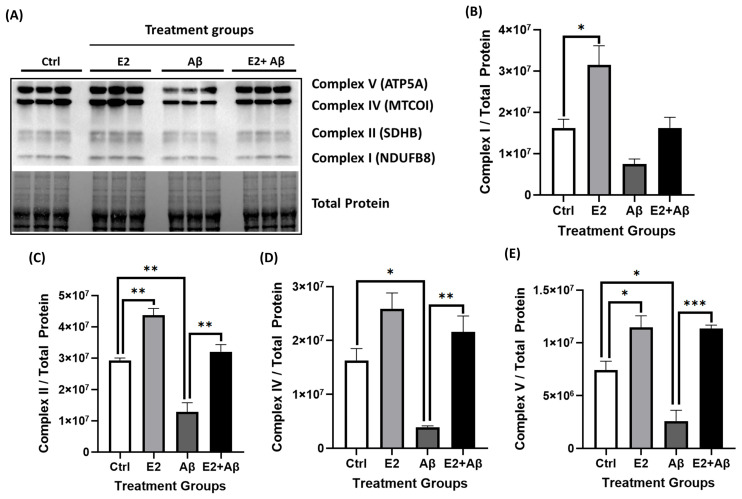
E2 pretreatment restores the Aβ-induced loss of mitochondrial electron transport chain (ETC) complexes. Primary cortical neurons were pretreated with 10 nM E2 for 48 h before exposure to 10 μM Aβ for 24 h. Mitochondrial ETC complex levels were assessed by Western blotting: (**A**–**E**) Complex I (NDUFB8), Complex II (SDHB), Complex IV (MTCO1), and Complex V (ATP5A) (normalized to total protein). Data are expressed as mean ± SEM, with n = 3 replicate cultures. Statistical significance was determined by one-way ANOVA with Tukey’s post hoc test; * *p* < 0.05, ** *p* < 0.01, and *** *p* < 0.001.

**Figure 6 ijms-26-06203-f006:**
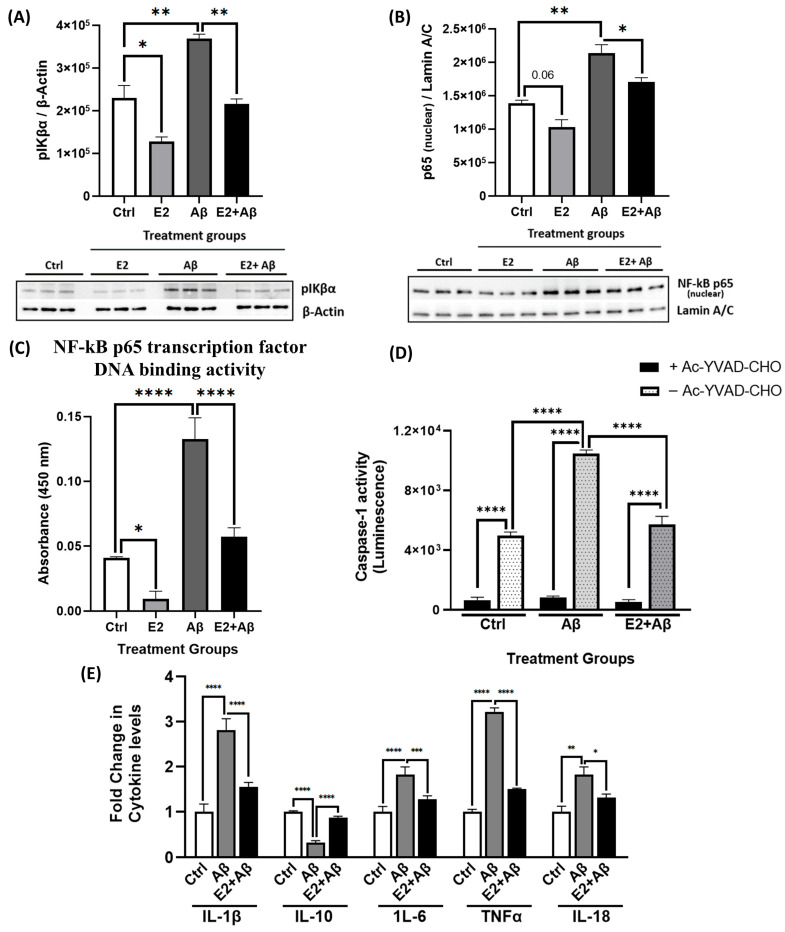
E2 pretreatment suppresses Aβ-induced NF-κB subunit expression, inflammasome activity, and pro-inflammatory cytokine production in primary cortical neurons. Primary cortical neurons were pretreated with 10 nM E2 for 48 h before exposure to 10 μM Aβ for 24 h. (**A**) pIκBα levels (normalized to β-actin) and (**B**) nuclear NF-κB p65 levels (normalized to Lamin A/C) were determined by Western blotting. (**C**) NF-κB activation was assessed by measuring the DNA binding activity of the NF-κB p65 subunit with respect to a specific dsDNA sequence containing the NF-κB response element, with data normalized to the total protein concentration in nuclear extracts. (**D**) Caspase-1 activity, a measure of inflammasome activation, was determined by a luminescence assay; the selective caspase-1 inhibitor Ac-YVAD-CHO was used to confirm specificity. (**E**) Pro-inflammatory (IL-1β, IL-6, TNFα, and IL-18) and anti-inflammatory (IL-10) cytokine levels in cell culture supernatants were measured by ELISA and reported as fold change relative to the control group. Data are expressed as mean ± SEM, with n = 3 replicate cultures for Western blot and NF-κB DNA binding assays (**A**–**C**) and n = 4 replicate cultures for caspase-1 activity and ELISA (**D**,**E**). Statistical significance was determined by one-way ANOVA with Tukey’s post hoc test; * *p* < 0.05, ** *p* < 0.01, *** *p* < 0.001, and **** *p* < 0.0001.

**Figure 7 ijms-26-06203-f007:**
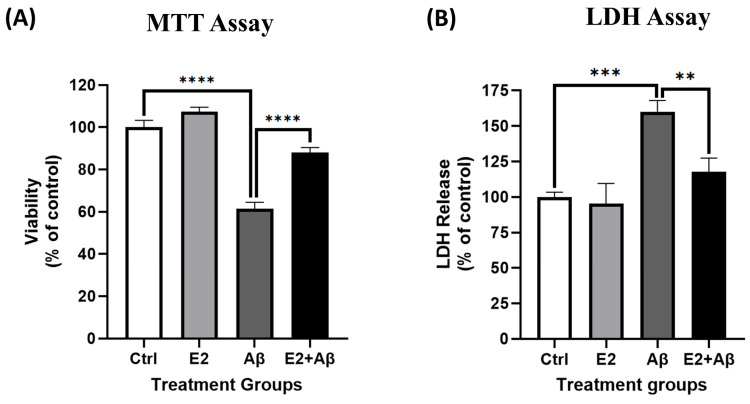
E2 pretreatment rescues Aβ-induced cytotoxicity and improves neuronal viability. Primary cortical neurons were pretreated with 10 nM E2 for 48 h before exposure to 10 μM Aβ for 48 h. (**A**) Cell viability was assessed using the MTT assay, and (**B**) cytotoxicity was measured using the LDH assay. Data are presented as a percentage of the control group and expressed as mean ± SEM, with n = 3–4 replicate cultures. Statistical significance was determined by one-way ANOVA followed by Tukey’s post hoc test; ** *p* < 0.01, *** *p* < 0.001, and **** *p* < 0.0001.

**Figure 8 ijms-26-06203-f008:**
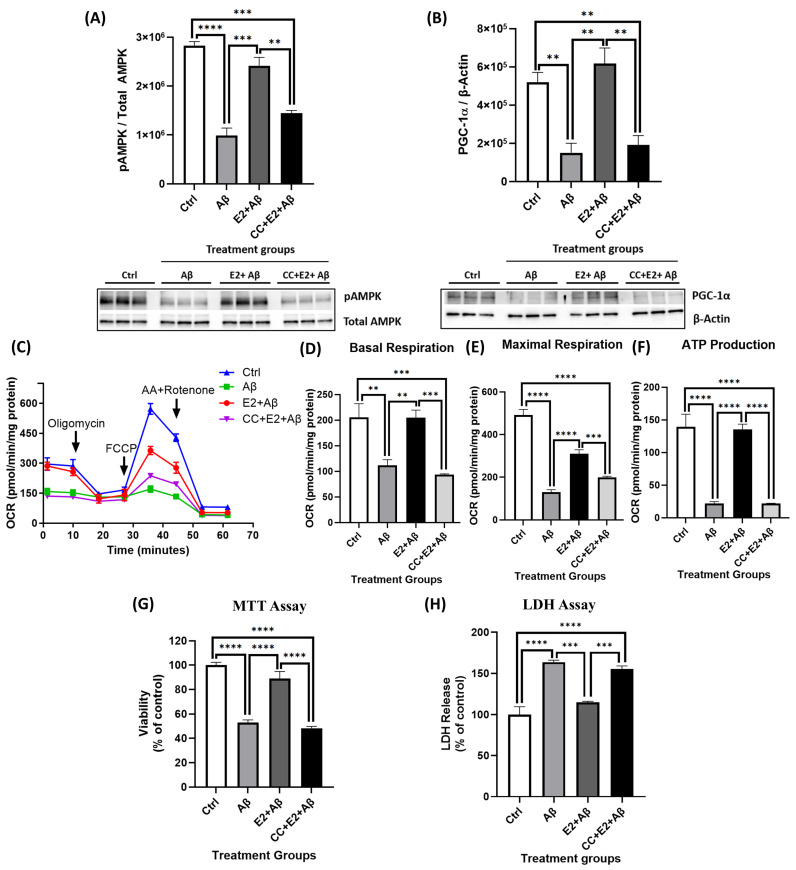
Blocking AMPK abolishes E2’s protective effects against Aβ-induced mitochondrial dysfunction and neurotoxicity. Primary cortical neurons were treated with 10 μM Aβ alone for 24 h, pretreated with 10 nM E2 for 48 h followed by Aβ, or pretreated with 5 μM Compound C (CC) for 1 h followed by E2 and Aβ. (**A**) pAMPK levels (normalized to total AMPK) and (**B**) PGC-1α levels (normalized to β-actin) were analyzed by Western blotting. (**C**–**F**) OCR, including basal respiration, maximal respiration, and ATP production, was measured via Seahorse XF24 Analyzer and normalized to protein concentration per well. (**G**) Cell viability was assessed by MTT assay, and (**H**) cytotoxicity was measured using the LDH assay, with data expressed as a percentage of the control group. Data are presented as mean ± SEM, with n = 3 replicate cultures for Western blot analyses (**A**,**B**), n = 5 replicate cultures for Seahorse assays (**C**–**F**), and n = 4 replicate cultures for MTT (**G**) and LDH (**H**) assays. Statistical significance was determined by one-way ANOVA with Tukey’s post hoc test; ** *p* < 0.01, *** *p* < 0.001, and **** *p* < 0.0001.

**Figure 9 ijms-26-06203-f009:**
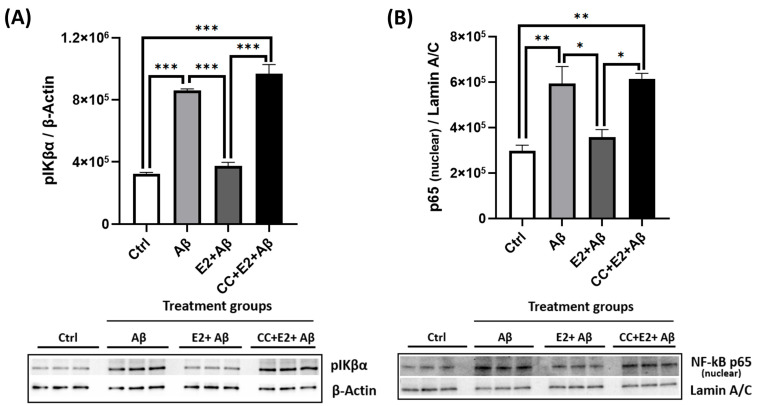
AMPK inhibition abolishes E2’s suppression of Aβ-induced NF-κB subunit expression and translocation. Primary cortical neurons were treated with 10 μM Aβ alone for 24 h, pretreated with 10 nM E2 for 48 h followed by Aβ, or pretreated with 5 μM Compound C (CC) for 1 h before E2 and Aβ. (**A**) pIκBα levels (normalized to β-actin) and (**B**) nuclear NF-κB p65 levels (normalized to Lamin A/C) were analyzed by Western blotting. Data are presented as mean ± SEM, with n = 3 replicate cultures. Statistical significance was determined by one-way ANOVA with Tukey’s post hoc test; * *p* < 0.05, ** *p* < 0.01, and *** *p* < 0.001.

**Figure 10 ijms-26-06203-f010:**
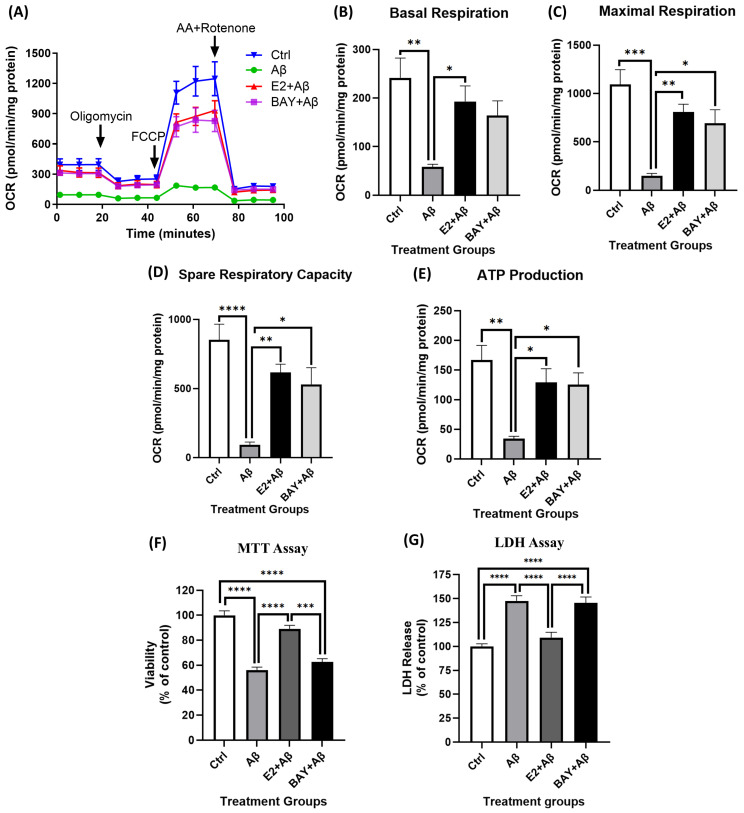
NF-κB inhibition rescues mitochondrial dysfunction but not Aβ-induced cytotoxicity. Primary cortical neurons were treated with 10 μM Aβ alone, pretreated with 10 nM E2 for 48 h followed by Aβ, or pretreated with 3 μM BAY 11-7082 (NF-κB inhibitor) for 1 h before Aβ exposure. (**A**–**E**) Mitochondrial OCR parameters, including basal respiration, maximal respiration, ATP production, and spare respiratory capacity, were measured using the Seahorse XF24 Analyzer after 24 h of Aβ exposure. Data were normalized to protein concentration per well and are presented as mean ± SEM, n = 5 replicate cultures. (**F**) Cell viability was assessed by MTT assays, and (**G**) cytotoxicity was measured using the LDH assay after 48 h of Aβ exposure, with data shown as a percentage of the control group and presented as mean ± SEM, n = 4 replicate cultures. Statistical significance was determined by one-way ANOVA followed by Tukey’s post hoc test; * *p* < 0.05, ** *p* < 0.01, *** *p* < 0.001, and **** *p* < 0.0001.

## Data Availability

Data will be made available upon request to the corresponding author.

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
