# Peer review of "Estradiol Prevents Amyloid Beta-Induced Mitochondrial Dysfunction and Neurotoxicity in Alzheimer’s Disease via AMPK-Dependent Suppression of NF-κB Signaling"

_ijms, 2025, doi:10.3390/ijms26136203_

Round 1

Reviewer 1 Report

Comments and Suggestions for Authors

The manuscript presents a well-designed and methodologically sound study investigating the role of estrogen signaling in neuroinflammation, with particular attention to the AMPK and NF-κB pathways. The authors use a combination of pharmacological inhibition and molecular analysis in cortical neurons to assess the effects of estrogen on Aβ-induced neuroinflammatory markers, which is both timely and relevant given the ongoing interest in neurodegenerative diseases like Alzheimer’s.

Comments:
1- While the introduction is informative, a brief mention of how these findings might relate to clinical implications or therapeutic angles would strengthen its impact.

2- In the Methods section, providing more detail on the source and characterization of Aβ (e.g., preparation method, aggregation status) would improve reproducibility.

3- Figure legends could benefit from stating sample size and statistical test details directly, even if this is also in the methods.

4- The English is generally very good, though a few minor grammatical edits could help improve flow in some sections.

Author Response

We sincerely thank the reviewer for their thoughtful and encouraging comments on our work. We truly appreciate their recognition of the study’s design and its relevance to ongoing research in neurodegenerative diseases.

Please find our detailed responses in the attached document.

Reviewer 2 Report

Comments and Suggestions for Authors

Excellent and important study linking estrogen, A-beta and AMPK.  The role of estrogen in AD has been debated for years---an important issue.   This study put AMPK as critical in the role of estrogen.  The findings are an essential to move beyond description and correlation.  The data and approach is elegant.

minor issues

abbreviations are not always spelled out on first usage--this would help the reader.

Author Response

We sincerely thank the reviewer for their positive and insightful feedback. We appreciate their recognition of the significance of our findings and their thorough review of the manuscript.

Please find our detailed responses in the attached document.
